# The Flavoproteome of the Model Plant *Arabidopsis thaliana*

**DOI:** 10.3390/ijms21155371

**Published:** 2020-07-28

**Authors:** Patrick Schall, Lucas Marutschke, Bernhard Grimm

**Affiliations:** Faculty of Life Science, Institute of Biology/Plant Physiology, Humboldt-University Berlin, Philippstraße 13 (Building 12), 10115 Berlin, Germany; patrick.schall@hu-berlin.de (P.S.); lucas.marutschke@hu-berlin.de (L.M.)

**Keywords:** riboflavin, flavoenzymes, flavin adenine dinucleotide, FAD, flavin adenine dinucleotide, FMN, flavocoenzyme, proteomics, gene families

## Abstract

Flavin mononucleotide (FMN) and flavin adenine dinucleotide (FAD) are essential cofactors for enzymes, which catalyze a broad spectrum of vital reactions. This paper intends to compile all potential FAD/FMN-binding proteins encoded by the genome of *Arabidopsis thaliana.* Several computational approaches were applied to group the entire flavoproteome according to (i) different catalytic reactions in enzyme classes, (ii) the localization in subcellular compartments, (iii) different protein families and subclasses, and (iv) their classification to structural properties. Subsequently, the physiological significance of several of the larger flavoprotein families was highlighted. It is conclusive that plants, such as *Arabidopsis thaliana*, use many flavoenzymes for plant-specific and pivotal metabolic activities during development and for signal transduction pathways in response to biotic and abiotic stress. Thereby, often two up to several homologous genes are found encoding proteins with high protein similarity. It is proposed that these gene families for flavoproteins reflect presumably their need for differential transcriptional control or the expression of similar proteins with modified flavin-binding properties or catalytic activities.

## 1. Introduction

Flavins are ubiquitous and essential for all living organisms [1]. Riboflavin as a precursor for flavin mononucleotide (FMN) and flavin adenine dinucleotide (FAD) has been described since more than 100 years [2]. In the 1930s, FMN and FAD have been discovered as essential cofactors for enzymes, which thereafter were also designated flavoenzymes [3,4,5]. Subsequently, riboflavin was classified as vitamin B2. While all organisms can synthesize FMN and FAD from riboflavin, only plants, fungi, archaea, and many bacteria are able to synthesize de novo riboflavin [1].

Research on the riboflavin biosynthetic pathway is documented for a long time and has been reviewed several times over the last two decades [1,6,7,8]. In brief, riboflavin is biosynthesized inside the plastids of plants by seven distinct enzymatic reactions [6]. Including the recently identified gene for the 5-amino-6-(5-phospho-D-ribitylamino) uracil phosphatase (AtPyrP2), at least one nuclear-encoded enzyme was described for each enzymatic step of the pathway [9]. One molecule of guanosine 5′-triphosphate (GTP) and two molecules of ribulose 5-phosphate serve as initial substrates [10,11], which are catalytically converted by the bifunctional *A. thaliana* RIBA1 protein comprising GTP cyclohydrolase II and 3,4-dihydroxy-2-butanone 4-phosphate synthase [12,13]. The GTP cyclohydrolase II activity was suggested as the rate limiting step in plant riboflavin biosynthesis [7,9].

Regulation of riboflavin biosynthesis has been studied in detail for bacteria, where all riboflavin biosynthesis genes are part of the *rib* operon [14]. One of the regulatory sequences in this operon codes for a riboswitch, which represses gene expression upon FMN-binding [15,16,17]. So far, a similar regulatory mechanism has not been found in plants [7]. *Beta vulgaris* and *Helianthus annuus* under iron depletion were reported to secrete a yellow pigment, which was identified as riboflavin 5′- and riboflavin 3′-sulphate [18,19]. Similarly, overexpression of the transcription factors *AtbHLH38* and *AtbHLH39* led to an up to 20-fold increased riboflavin content in tobacco, but not in Arabidopsis [20]. Having said that, it is not very much known about the transcriptional and posttranslational control of the synthesis of riboflavin in plants.

Riboflavin is not used as a catalytic cofactor but is the precursor for the formation of flavocoenzymes. Riboflavin kinase (RK) phosphorylates riboflavin to FMN, which is converted to FAD by FAD synthetase (FADS). The backreactions are catalyzed by FAD- and FMN- hydrolases (FADHy, FMNHy) [21]. In prokaryotes, bifunctional RK/FADS are found apart from monofunctional enzymes [22,23,24]. In contrast, mammals, yeasts, and plants exclusively use separate enzymes with RK and FADS activity [21,25,26,27].

While plant enzymes involved in riboflavin biosynthesis show an exclusive localization in plastids, the enzymes for flavocoenzyme synthesis and degradation are localized in different cellular compartments, the cytosol, the plastids and, conceivably, the mitochondria [8,21]. The plastidal enzymes for the flavocoenzyme metabolism and their corresponding genes are already reported except the RK, of which hitherto only an activity was described in pea chloroplasts [21]. A bifunctional RK/FMNHy acts in the cytoplasm, while a cytoplasmic FADS is only predicted [28]. Flavin synthesis and degradation were determined in the mitochondria, but enzymes were not identified, yet [21,29]. Beyond that, no other cellular compartment was demonstrated to contribute to flavin synthesis in plants. This is in contrast to the nucleus-localized FADS activity in mammals [30,31]. 

Regulation of the flavocoenzyme synthesis as well as degradation and transport of flavins are barely understood in plants. A first experimental indication for translocation of riboflavin between cytoplasm and mitochondria was reported in addition to the export of FAD from the mitochondria. The export of one of the three flavins from plastids has not been observed, but cannot be excluded, as riboflavin is exclusively biosynthesized in plastids [8,21]. Thus, studies on the metabolism of riboflavin and flavins still meet many challenges in the near future. 

Many cellular processes directly depend on flavin biosynthesis, as flavoenzymes are catalyzing a broad spectrum of different reactions [32,33]. For example, they are involved in apoptosis, photochemistry, redox regulation, chromatin organization, DNA repair, protein folding, and energy metabolism [32]. Even though there are transferases, lyases, isomerases, and ligases, the majority of flavoenzymes are catalyzing redox reactions [33]. This observation is not unexpected, considering the chemical properties of the flavins. Due to the isoalloxazine ring, they have three different redox states, allowing them to transfer either one or two electrons [34].

We would like to bring the flavoproteome of the model plant *Arabidopsis thaliana* to the readers’ attention and describe here the first plant flavoproteome. We highlight general aspects of this proteome and compare our data with published flavoproteomes of yeast and humans. We subsequently highlight the physiological significance of several of the larger flavoprotein families.

## 2. Results-General Aspects of the Flavoproteome of *Arabidopsis thaliana*

### 2.1. Flavoproteins, Localization, and FMN/FAD Distribution

First, we created a list of genes coding for potential flavoenzymes in *A. thaliana* by searching in the gene ontology (GO) database, the UniProt database, and the Protein Data Bank (PDB). In addition, we verified every gene locus from this preliminary list with available references in the literature (Appendix A). Thereby, some missing flavoenzyme-encoding genes were identified and additionally listed. Consequently, three different evidences for all genes listed were applied to indicate how the annotation as a flavoprotein is supported. Evidence code one classifies genes, which encode proteins containing an experimentally proven flavin-binding site. This was achieved by chromatography analyses of flavins or crystal structure analysis of the holoprotein. The second code includes genes coding for proteins to which flavocoenzymes were assigned in the literature, mainly by sequence homology, but without experimental evidence. Finally, genes were added to a third evidence code, which are predicted to be flavoenzymes by sequence analysis, but either without any reference in the literature on flavin-binding or any citation at all (Appendix A). 

Ultimately, our list includes 249 genes encoding potential flavoproteins. There are 47 genes belonging to the criteria of evidence code one, 161 to evidence code two, and 31 to the evidence code three. Derived from a total number of 27,416 protein-encoding genes in Arabidopsis (TAIR10), the entire genome contains in total about 0.91% of genes for flavoproteins. This magnitude of genes is in the range between the predicted amount of flavoprotein-encoding genes of the *Mycoplasma genitalium* (~0.1%) and *Mycobacterium tuberculosis* (~3.5%) genome [33]. With 249 entries, the flavoproteome of Arabidopsis is larger than the recently published human and yeast flavoproteomes consisting of 90 and 68 flavin-dependent proteins, respectively. These numbers account for 0.45% of the total human proteome, when 19.823 protein-encoding genes are considered, and 1.1% of the yeast flavoproteome in relation to their total proteome [35,36]. 

Among the 249 flavoproteins, 211 and 32 proteins exclusively bind FAD and FMN, respectively (Figure 1a). A strong preference towards FAD seems to be a general aspect of all flavoproteomes [33]. With about 85% FAD-binding flavoenzymes, the relative FAD usage in Arabidopsis is very similar compared to that of the total human flavoproteome, which consists of 84% FAD-dependent enzymes [36]. In addition, we identified six flavoenzymes, which are annotated to bind both flavocoenzymes—three cytochrome P450 reductases and three glutamate synthases. The FMN-binding of glutamate synthases was reported [37,38], while FAD-binding was just predicted. We decided to keep the Arabidopsis glutamate synthases to the list of FAD-dependent flavoenzymes, as orthologous proteins were shown to bind both FMN and FAD [39,40]. 

We also grouped the flavoenzymes to subcellular compartments (Figure 1b). Information in published manuscripts about the subcellular localization was found for 100 flavoproteins (~ 40%). The subcellular sites of the remaining flavoproteins were assessed by GO terms and Uniprot annotations. We succeeded to define the cellular compartments for additional 141 enzymes (~ 56%), while the localization of eight entries could not be predicted. Among the subcellular sites plastids contain 61 flavoenzymes, followed by the cytoplasm with 57, and mitochondria with 49 proteins. We also identified 42 additional potentially membrane-associated flavoenzymes (i.e., without the candidates associated to plastidal and mitochondrial membranes). In addition, 38 proteins were assigned to the cell wall or extracellular space, 37 to the nucleus, 23 to the peroxisomes, 10 to the endoplasmatic reticulum (ER), and 16 to other subcellular sites.

Recently, an updated list of 510 EMBRYO-DEFECTIVE (EMB) genes was presented [41], which contains several flavoenzyme-encoding genes. We detected six flavoenzyme-encoding genes (AT1G24340, AT1G48850, AT1G49880, AT3G02280, AT3G54660, AT5G14760), which are assigned to be essential for embryo development [42,43,44,45,46]. AT1G24340 is predicted to encode a FAD-dependent flavoenzyme (see the third evidence code). Considering that the list of EMB genes is not complete (as 750–1000 EMB genes are estimated in Arabidopsis [41]), it is assumed that more flavoenzymes are essential for embryo development. In continuation, a list of embryo-lethal double mutants (embryo lethality only in combination of two mutated genes) was published. This list is much shorter (83 combinations) and contains four flavoenzyme-encoding genes. One pair of genes consists of the HAL3A (AT3G18030) and HAL3B (AT1G48605) genes encoding phosphopantothenoyl-cysteine decarboxylases [47]. The other pair is composed of the squalene monooxygenases SQE1 (AT1G58440) and SQE3 (AT4G37760) [48]. The occurrence of flavoenzyme-encoding genes in the two compilations of the EMB genes points to two different aspects. It refers firstly to the potential importance of flavoenzymes for embryogenesis and essential functions during development and secondly to gene duplications and functional redundancies among the listed flavoenzymes (see the discussion). Accordingly, it is expected that the combination of further knock-out mutants with deficiency of other homologous genes will likely cause an EMB phenotype.

### 2.2. Flavoprotein Families in Arabidopsis thaliana

According to the Enzyme Commission number (EC number) the numerical classification system reveals that 88.3% of the flavoproteins belong to oxidoreductases (Figure 2). These findings are in accordance with the redox properties of the isoalloxazine ring of FMN and FAD. The second and third largest groups of flavoenzymes are lyases and transferases with 3.2% and 1.2% of the total flavoproteome, respectively. A significant number of 6.83% of all flavoproteins could not be classified to any enzymatic reaction pointing to the need of more elaborating studies on flavoproteins in the future.

The further classification of the coenzymes in subgroups revealed that the largest subgroup with EC number 1.1 contains 52 entries. These enzymes act on the CH-OH group of the donor substrate. The second and third largest subgroups are EC 1.14. and EC 1.3. with 28 and 27 entries, respectively. EC number 1.14. is categorized as “acting on paired donors, with incorporation or reduction of molecular oxygen” and EC number 1.3. as “acting on the CH-CH group of donors” [49]. This list of classified enzymes indicates the importance of flavoproteins for a small but rather pronounced variety of enzymatic reactions in *Arabidopsis thaliana* (Figure 2).

In order to emphasize the importance of flavoproteins we also arranged all flavoproteins to protein families and their associated subclasses in an order of descending members per family (Figure 3). Interestingly, two flavoprotein families are strikingly overrepresented among all flavoproteins. The oxygen-dependent FAD-linked oxidoreductase family is most abundant with 41 different gene products (AGI loci), followed by the flavin-containing monooxygenase family with 29 entries (Figure 3a).

The FAD-linked oxidoreductase family consists mainly of enzymes from the berberine bridge enzyme (BBE) family with 27 single entries (Figure 3b). The flavin-containing monooxygenase (FMO) family consists mainly of two enzyme subclasses: Indole 3-pyruvate monooxygenases (11 entries) and flavin-monooxygenase glucosinolate S-oxygenases (16 entries). Furthermore, an additional protein family with high abundance in the flavoproteome is the subclass of respiratory burst oxidase homolog (RBOH) proteins. The specific functions of these enzyme classes are discussed further below.

### 2.3. Structural Analysis of Flavin-Binding Domains 

A search for structural clans among the flavoproteome of *A. thaliana* in the Pfam database revealed a strong tendency for the TIM barrel domain in FMN-binding proteins (22 entries) and for a Rossmann fold in FAD-binding proteins (Figure 4) [50]. These results are in accordance with findings of Macheroux et al. (2011), who described the TIM barrel motif and the Rossmann fold to be the main structural domains for binding of FMN and FAD in flavoproteins, respectively [33].

### 2.4. Flavoprotein Functions in Biological Processes

To elucidate the involvement of flavoproteins in different biological processes we applied a gene ontology enrichment analysis (GO enrichment analysis) (Figure 5). The GO enrichment analysis indicates that most of the flavoproteins are involved in the oxidation-reduction process. Furthermore, it turned out that a majority of candidates of the flavoproteome contributes to the phytohormone metabolism. These findings correlate with results depicted in Figure 2 and Figure 3 confirming a huge portion of the flavoproteome to be associated with enzyme classifications, which act on CHO groups (Figure 2). Additionally, other flavoproteins participate in the metabolism of auxin (Figure 3, indole-3-pyruvate monooxygenase) and cytokinin (Figure 3, cytokinin dehydrogenase). Interestingly, a significant number of flavoproteins plays also a role in the primary and secondary metabolism for carbon, nitrogen or sulphur assimilation as well as in the response towards abiotic stress (e.g., wounding, cadmium, pollution, reactive oxygen species) and amine metabolism, which is also often related to the plant response against pathogens (Appendix A) [51,52,53]. 

## 3. Discussion—Function of Flavoproteins in the Carboxylic Acid Metabolism

### 3.1. Berberine Bridge Enzymes

The BBE-like proteins consist of a group of enzymes belonging to the FAD-linked oxidases [54]. All BBE-like proteins share a common vanillyl alcohol-oxidase fold. This enzyme group inherited its name from the conversion of (S)-reticuline to (S)-scoulerine by a (S)-reticuline oxidase, which is a BBE in *Eschscholzia californica* [55]. The (S)-reticuline oxidase closes a ring by forming a C-C bond, which is named the “berberine bridge” [55]. BBE-like proteins have been found in bacteria, fungi, and plants [54].

It has been shown that the covalently bound FAD has an increased redox potential compared to a free flavin [56]. The family of BBE-like proteins is involved in a variety of enzymatic reactions with two-electron and four-electron oxidation reactions [54]. In plants, BBE-like proteins are ubiquitous and are involved in many different enzymatic reactions such as oxidization of alcohols and carbohydrates [54]. In *A. thaliana (At),* 28 BBE-like proteins have been found [57]: 27 BBE-like proteins, each with an individual gene locus (Appendix A), and the *At*BBE-like 19 encoding gene forming two splice variants (AT4G20830.1 and AT4G20830.2). It has been shown that BBE-like 13 as well as *At*BBE-like 15 are both located in the cell wall and have monolignol oxidoreductases activity [58]. Both enzymes catalyze the oxidation of sinapyl-, coniferyl-, and p-coumaryl alcohol to their aldehyde forms (Figure 6) [58]. Those alcohols (monolignols) and their aldehyde forms are the building blogs of lignin biosynthesis [58]. Furthermore, it could be shown that 12 other AtBBE-like proteins have a high sequence similarity to AtBBE-like 13 and AtBBE-like 15 indicating a major role of AtBBE-like proteins in cell wall lignification [58]. 

In addition, gene expression studies revealed a transcript accumulation for AtBBE-like proteins 3–7 after exposure to pathogen and osmotic stress reflecting a role in the response to environmental stresses [58,59]. It could be shown that AtBBE-like 3 is involved in the synthesis of the cyanogenic compound 4-hydroxyindole-3-carbonyl nitrile and catalyzes indole cyanohydrin to indole-3-carbonyl cyanide/nitrile (Figure 6) [59]. The crystal structure of AtBBE-like 28 revealed a distinct active site only found in the family of Brassicaceae [60]. AtBBE-like 28 shows only a mono-covalent bond of FAD to the enzyme through the side chain of histidine 111, while AtBBE-like 15 as most of BBEs proteins has a bi-covalent attachment of the isoalloxazine of FAD to cysteine 176 and histidine 115 of the protein (Figure 7) [58,60]. A T-DNA insertion line of the gene encoding AtBBE-like 28 was characterized by less biomass and a minor salt stress tolerance indicating that despite the huge protein family and the structural similarity of many family members the specific function of AtBBE-like 28 is not or only partially compensated by one of the other AtBBE-like representatives [60]. 

### 3.2. Flavin Monooxygenase Glucosinolate S-Oxygenase Enzymes

Flavin monooxygenase glucosinolate s-oxygenase enzymes (Figure 3) catalyzes S-oxygenation of methylthioalkyl to methylsulfinylalkyl in the biosynthesis of glucosinolates (GSLs) (Figure 8) [62]. GSLs are sulphur containing secondary metabolites in the plant family of Brassicaceae with various functions depending on their side chain modification [63,64].

GSLs play a role in the plant defense, when endogenous thioglucosidases (for example after wounding) break GSLs down into their biological active compounds, such as isothiocyanates, thiocyanates, and nitriles [65,66,67]. Furthermore, it was shown that isothiocyanates have protective properties in different human cell tumors [68,69,70,71]. In *A. thaliana*, seven FMO_GS-OX_ isoforms have been described in the literature so far [62,64,72]. Our search for flavoproteins identified nine additional putative FMO_GS-OX-_like proteins by sequence similarity. Biochemical analysis of FMO_GS-OX2,3,4,5_ has shown that FMO_GS-OX5_ specifically forms long-chain 8-methylthiooctyl GSLs, while FMO_GS-OX2,3,4_ have a broader substrate specificity, without any preference for molecules with varying chain lengths [64]. In addition, analysis of Arabidopsis gene expression after treatment with different hormones and upon temperature stress revealed a differential sensitivity of the seven described FMO_GS-OXs_ towards different environmental stimuli suggesting a specific role of FMO_GS-OX_ variants in the biosynthesis of GSLs [72]. 

### 3.3. Flavoproteins Function in Hormone Metabolism

The GO term enrichment (Figure 3) and the enzymatic classification (Figure 2) highlight flavoproteins in the phytohormone metabolism. The two most abundant subclasses of protein families are cytokinin dehydrogenases and indole-3-pyruvate monooxygenases (designated YUCCA proteins) with seven and eleven entries, respectively [73]. YUCCA catalyzes the conversion from indol-3-pyruvate to indol-3-acetic acid at the rate limiting step of tryptophan-dependent auxin biosynthesis in *Arabidopsis thaliana* (Figure 9) [74]. The expression of different *YUCCA* genes is spatio-temporally controlled indicating their distinct functions. However, only triple and quadruple *Arabidopsis thaliana* knock-out mutants of different *YUCCA* genes reveal a severe impact on plant growth and the reproduction rate [75,76]. Notably, overexpression lines of YUCCA1 facilitate a 50% increased accumulation of free IAA compared to the wild type [77].

Apart from the contribution of YUCCAs to auxin synthesis, other families for flavoproteins are also associated with plant hormone metabolism. The biosynthesis of abscisic acid (ABA) depends on two different flavoproteins: The FAD-dependent zeaxanthin epoxidase and the FAD-binding aldehyde oxidase (AAO3) [78]. The biosynthesis of jasmonic acid also requires at least one flavoenzyme, the FMN-dependent 12-oxophytodienoate reductase (OPR3) [79]. A role of the FAD-binding peroxisomal acyl-coenzyme A oxidase 1 (ACX1) in jasmonic acid biosynthesis was suggested, but not demonstrated so far [80].

Additionally, the cytokinin oxidases/dehydrogenase (CKX)-family are responsible for inactivation of the cytokinin by oxidative cleavage of the side chain [81,82]. Cytokinins are a class of N^6^-substitued purine derivates, which are involved in the regulation of cell division and many other developmental events [83]. The seven CKXs in Arabidopsis (CKX1-CKX7) show an amino acid identity between 34.3% and 65.9%, while the FAD-binding domain is well conserved among all of them [82]. The covalently bound FAD coenzyme has first been observed in maize CKX1 and was also confirmed in a crystal structure of AtCKX7 [84,85]. Over the years, the physiological reasons for the occurrence of seven CKX proteins have been discussed. One main difference among the CKX proteins is their subcellular localization. Targeting to the ER and secretion to the extracellular space was confirmed for CKX2 and predicted for CKX4, CKX5, and CKX6 [82,86]. CKX1 and CKX3 were demonstrated to be localized in the vacuole, while CKX7 was shown to be targeted to the cytosol [86,87]. Furthermore, CKXs not only differ in their subcellular localization, but also possess different catalytic properties, including the substrate specificity and pH dependency [82,88,89].

Considering the relevance of cytokinin and its regulation for plant development, a possible application of modified cytokinin metabolism was discussed in crop plants [82,90]. It was reported that reduced expression of OsCKX2 leads to an increased number of reproductive organs and therefore enhances rice grain yield [91]. Similar results have been shown by silencing the *HvCKX1* gene in barley [92]. In another interesting approach *CKX* genes were overexpressed in tobacco resulting in enhanced drought and heat stress tolerance [93,94].

### 3.4. Flavoproteins Function in the Formation of Reactive Oxygen Species (ROS) 

Another family of flavoenzymes in *A. thaliana* contains ten RBOH members. RBOHs are homologous to the mammalian FAD-dependent, membrane-bound NADPH oxidase gp91*^phox^*, which produces ROS [95,96]. In contrast to their mammalian homologues, plant RBOHs contain an N-terminal regulatory region with peptide motifs for Ca-binding, protein-binding, and phosphorylation [95,97]. This feature makes RBOHs unique among other ROS-producing enzymes as they serve in ROS-mediated signaling [97]. 

The function of superoxide produced by NADPH oxidases in plants is associated with a broad spectrum of different stress responses [98]. The different biotic stress syndromes, including the extracellular interaction with plant pathogens, are always mediated by ROS production. Apart from their toxic and damaging impact on microbial pathogens [99], ROS also act as secondary messengers to induce an intracellular defense response. Plants exposed to bacterial phytopathogens exhibit higher *RBOHD* transcript levels leading ultimately to ROS accumulation, which induces a global defense response [100]. Moreover, it has been shown that the recognition of bacterial flagellin by the receptor FLS2 leads to ROS production through NADPH oxidases, which subsequently promote stomatal closure to prevent further invasion of pathogens into the apoplast space [101]. However, this stomatal immunity due to ROS production of NADPH oxidases can also be induced by ABA signaling [102]. 

NADPH oxidases are also involved in many abiotic stress responses. A *rbohd* x *rbohf* double mutant shows a perturbed growth phenotype under salt stress [103]. In the same line, the *RBOHI* overexpressor line exhibits enhanced drought tolerance [104]. A further abiotic stress response, which includes the role of NADPH oxidases, is hypoxic stress. The inactivation of the *RBOHD* or *RBOHF* genes leads to a hypoxia-sensitive phenotype [105]. In accordance to this observation, the transcript levels of several *RBOH* genes increase under hypoxic stress [106]. Furthermore, ROS accumulation under heavy metal (copper and cadmium) stress was also suggested to be associated with the NADPH oxidase activity [107]. Beyond that, NADPH oxidases are involved in cellular response to temperature stress, including heat stress [108,109]. 

The impact of NADPH oxidases in many different stress signaling responses might be the reason for the emergence of up to ten *RBOH* genes in Arabidopsis. Based on differential expression analysis of these *RBOH* genes it was proposed that specific stress factors correspond to specific induction and activation of individual members of this protein family, rather than their tissue and cell type-specific expression [97]. 

Other FAD-dependent enzymes involved in ROS production are the polyamine oxidases described in maize, tobacco, and other plants [52,110]. In Arabidopsis five isoenzymes of polyamine oxidases (PAO1-PAO5) are known, which catalyze polyamine re-conversion under production of H_2_O_2_ [111].

## 4. Conclusions

This manuscript contains a first collection of FAD- and FMN-binding proteins in *Arabidopsis thaliana* and is consistent with previous efforts to compile a flavoproteome of other Eukaryotes, e.g., yeast and humans [35,36]. The flavoproteome analysis revealed (i) which catalytic reactions are mainly performed by flavin-dependent enzymes in Arabidopsis, (ii) in which cellular compartments these proteins are mainly located, and (iii) to which cellular processes these proteins are contributing. It is apparently compelling that plants, such as *Arabidopsis thaliana*, use many flavoenzymes for plant-specific and pivotal metabolic activities during development or endogenous and environmental signaling. In doing so, it is striking that often two up to several homologous genes contribute to a single flavin-dependent enzymatic step.

We are aware, that this first survey of flavin-binding proteins is most likely far from being complete. Only when functional analyses of unknown proteins will become more complete and tend to cover the entire proteome of one plant species, the real number of flavin-dependent enzymes can be sufficiently assessed. Moreover, as only a few flavoproteins are structurally elucidated and only a few structures of putative flavin-binding proteins were inferred from known structures, it is presently rather vague to suggest common structural properties of peptide domains for FMN- and FAD-binding. However, the Rossmann fold has been identified in several flavin-dependent enzymes and is evolutionarily one of the oldest and most conserved domains [112]. Based on alignments of homologous sequences secondary structures are predicted which may assist in assigning proteins to structurally related protein clans among the FAD- and FMN-binding proteins of the flavoproteome. 

It might be of interest to assess the number of gene duplications and overall small and larger gene families encoding flavoenzymes with a high sequence similarity. The analysis of the flavogenome revealed sets of genes, which comprises families of two up to many representatives (Appendix A, Appendix A). Provided that the threshold level for homologous gene products is set at 60%, 50%, and 30% of sequence identity, in total 107, 143, and 173 flavoproteins were collected in this list (Appendix A). Appendix A comprises all flavoprotein-encoded genes and the number of gene products, which share sequence identity of at least 30%, 50%, and 60 %, respectively. Based on these three different classifications of sequence identities, the gene family with the highest number of redundant gene copies consists of maximal 26, 12, or 7 representatives, while at least 53, 75, and 76 pairs of homologous genes were found in the list of structurally related flavoproteins with 60%, 50%, and 30% sequence identity, respectively. These gene copies were found either in tandem in close proximity or distributed over the entire genome. It is not predictable without further biochemical analysis whether these proteins found with a different grade of sequence similarity are still functioning for the same cellular processes. On the other hand, it is not excluded that the members of these protein families with sequence similarities share similar or identical functions, independently whether a group of duplicated genes or a group of multiple representatives exist. As seen for families of flavoproteins contributing to the metabolism of phytohormones and for signal transduction pathways in response to biotic and abiotic stress, flavoenzymes encoded by several different genes in tendency act on important regulatory steps of plant specific cellular processes. As a result, redundancy of genes for flavoproteins is presumably explained by a plant-specific role of the respective family of gene products during development or/and for essential metabolic processes specifically required in certain tissues or cell-types or during stress conditions. Thus, homologous genes as a gene tandem or distributed on different chromosomes facilitate either a different transcriptional control or the expression of similar proteins with modified flavin-binding properties or catalytic activities. In this context, it is proposed that frequently changing redox states and substantial offensives of oxidative stress conditions in plants require a defined subset of flavin-dependent enzymes with a varying redox potential. In summary, the gene families with different members would at least increase the metabolic potency and improve versatility to changing stressors. 

Ultimately with reference to the families of structurally related flavoproteins, it is stated that more elaborated bioinformatic analyses are required to assign the origin of each homologous gene either to the genome of the host cell or the endosymbionts. In conclusion, considering the high number of proteins with high sequence similarity in the flavoproteome it becomes obvious that the proteome contains many members with redundant functions and the adequate number of structurally unrelated flavoproteins with specific function is much lower than the currently detected 249 representatives of FMN- and FAD-binding proteins.

In the end there are still many flavoproteins with unknown functions. Additionally, in the same line, there is still very little known about flavin storage proteins or riboflavin and FAD/FMN-specific transporters. As riboflavin synthesis exclusively occurs in plastids and flavoproteins are found in all cellular compartments, many specific transporters for riboflavin/FMN/FAD are predicted still to be identified in the plant proteome. The current annotations of genes in plant genomes do not provide further information on proteins with these functions. When more intensive structural analysis of membrane-bound proteins will support future predictions on flavin-binding and channeling, putative candidate proteins can be identified. 

Last but not least, it is hypothesized that mutants for genes involved in flavin synthesis or transport will not automatically show a specific flavin-type phenotype. It is at present questionable, whether a flavin-specific mutant phenotype could be visualized upon the deficiency of riboflavin or FAD/FMN. It is conceivable that the deficiency of flavins can either occur exclusively in one of the cellular compartments, in which a deficient flavoprotein is localized, or affect multiple cellular processes causing a pleiotropic phenotype. Therefore, a screen for mutants with a shortcoming in flavin synthesis and metabolism or of certain flavin-binding proteins, for example by a forward genetics approach, is not automatically promising, as the perturbation of the subcellular homeostasis of FAD and FMN could either affect many different flavin-dependent enzymes acting on too many diverse processes in specific cellular compartments, or is not distinguishable from WT control plants due to a putative intracellular possibility to exchange of FAD/FMN among the cellular compartments. Moreover, as indicated above, a flavin-dependent mutant phenotype might not be detectable when more than one gene copy for a flavoenzyme exists and the other gene product(s) facilitate(s) compensation of the lack of mutated gene. 

## 5. Experimental Methods

### 5.1. Screening of Databases for Flavoproteins

The gene ontology (GO) resource database was screened for putative flavoproteins with a Python script extracting gene loci associated with the GO terms (molecular function) “FAD binding”, “flavin adenine dinucleotide binding” or “FMN binding”. Note that the GO term “flavin mononucleotide binding” does not exist in the GO database. Furthermore, the Protein Data Bank was screened for plant protein structures containing FMN or FAD [113]. Subsequently, the genes in a preliminary list were verified through cited references for flavoproteins and the UniProt database [114].

### 5.2. Screening for Structural Clans in Flavoproteins

In order to identify structural clans in the flavoproteome of *A. thaliana* all protein sequences from Appendix A have been analyzed with profile hidden Markov Models in the Pfam database (http://pfam.xfam.org/) [50].

### 5.3. Gene Ontology Enrichment and Visualization

AGI numbers from Appendix A were used to generate a GO enrichment with the agriGO (version 2.0) webserver (http://systemsbiology.cau.edu.cn/agriGOv2) [115]. The GO enrichment was generated with a p-value cut off p < 0.05 and the TAIR genome locus (TAIR10 2017) database as the background/reference genome [116]. The visualization was done with the REVIGO webserver (http://revigo.irb.hr) together with the R programming language in R-Studio (RStudio PBC, Boston, USA [117].

### 5.4. Software and Webtools

Chemical structures were drawn with ChemDraw 19.1 (PerkinElmer, Waltham, USA).

PyCharm (JetBrains, Prague, Czech Republic) was used to search for flavoenzymes in the GO dataset (provided by The Arabidopsis Information Resource, https://www.arabidopsis.org).

Microsoft Excel (Microsoft Corporation, Redmond, USA) was used for chart creation.

Microsoft Word (Microsoft Corporation, Redmond, USA) was used for preparation of the manuscript.

AgriGo V2 (http://systemsbiology.cau.edu.cn/agriGOv2/).

Statistical test method: Fisher, Gene ontology type: Complete go.

Significance level: 0.05, Minimum number of mapping entries: 5. 

Phyre 2 web server (http://www.sbg.bio.ic.ac.uk/phyre2/html/page.cgi?id=index) was used to predict the 3D structure of unknown proteins [118].

## Figures and Tables

**Figure 1 ijms-21-05371-f001:**
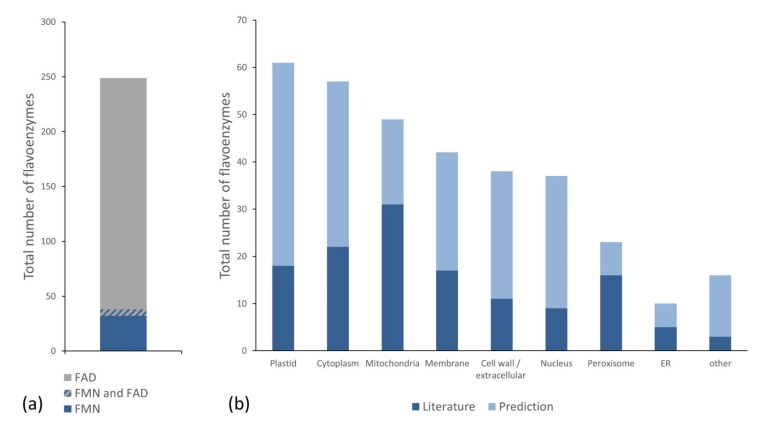
General statistics of the flavoproteome. (**a**) Shows the distribution of the flavocoenzymes flavin mononucleotide (FMN) and flavin adenine dinucleotide (FAD) among the flavoenzymes. (**b**) Shows the number of flavoenzymes, targeted to the indicated localization. Localizations described in the literature are represented in dark blue. For other flavoenzymes, we used gene ontology (GO) and UniProt annotations to predict the localization (pale blue).

**Figure 2 ijms-21-05371-f002:**
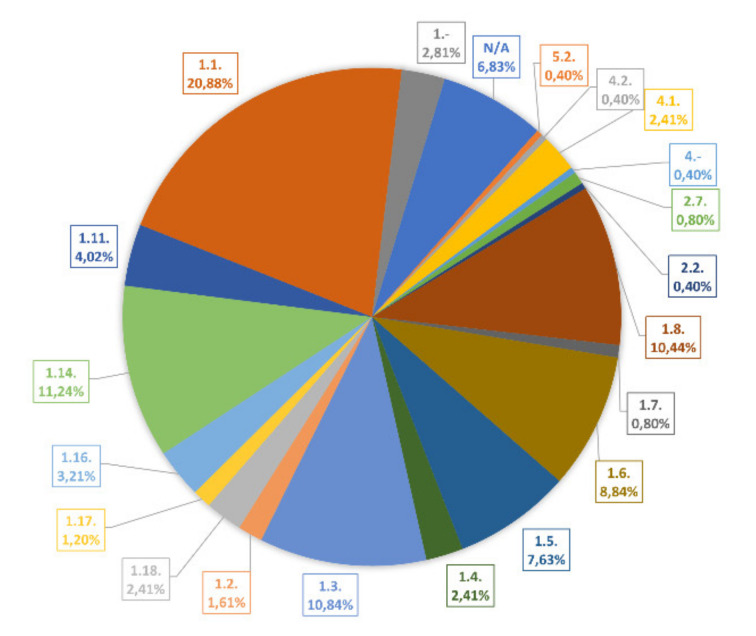
Distribution of the different enzyme classes among the flavoproteome of *A. thaliana*. The pie chart was generated by the enzyme classification according to the Enzyme Commission numbers (EC number) as summarized in Appendix A: 1. Oxidoreductases; 2. transferases; 4. lyases; 5. isomerases; N/A: Not available. The assignments to the enzyme classes and enzyme subclasses are found in https://www.brenda-enzymes.org/ecexplorer.php?browser=1. All Arabidopsis flavoproteins are depicted in Appendix A.

**Figure 3 ijms-21-05371-f003:**
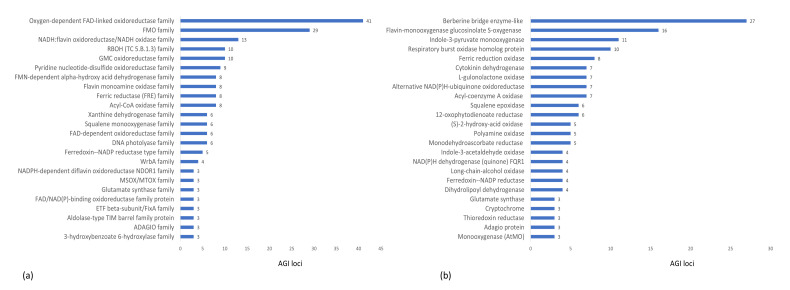
(**a**) Accumulation of protein families and (**b**) their subclasses among the flavoproteome in *A. thaliana*. Chart (**a**) and (**b**) are based on the AGI locus numbers from Appendix A. Protein families and subclasses with an accumulation of less than three AGI loci have not been considered in these charts.

**Figure 4 ijms-21-05371-f004:**
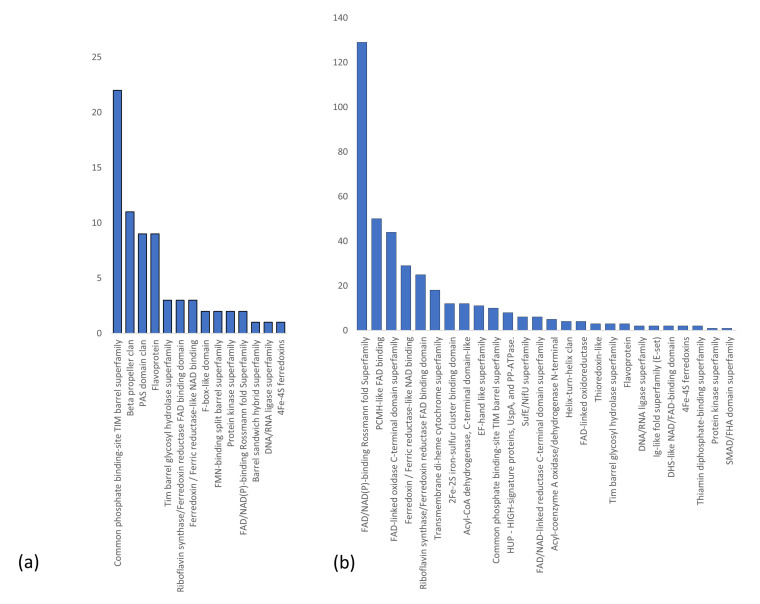
Distribution of structural clans in (**a**) FMN- and (**b**) FAD-binding proteins. The structure clans of all 249 proteins from Appendix A have been analyzed with hidden Markov models and the Pfam database.

**Figure 5 ijms-21-05371-f005:**
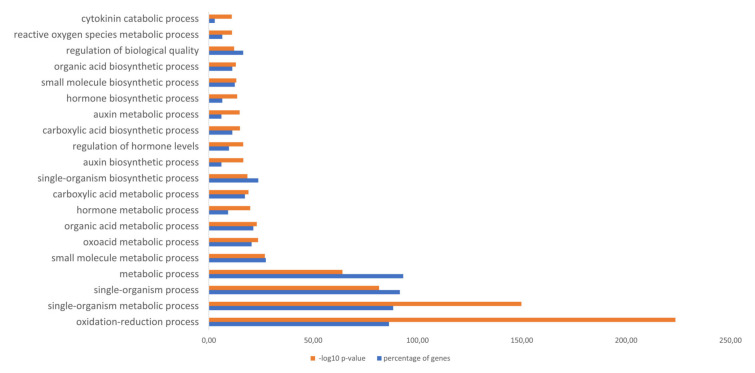
GO term enrichment of the flavoproteome of *Arabidopsis thaliana*. Orange represents the 20 highest log10 p-values of the enriched AGI numbers from Appendix A tested against the background database The Arabidopsis Information Resource (TAIR) genome locus (TAIR10_2017). The blue labeling represents the percentage of genes derived from all of the 249 genes of Appendix A. The p-value cut off is *p* < 0.05. The complete GO term enrichment can be found in Appendix A.

**Figure 6 ijms-21-05371-f006:**
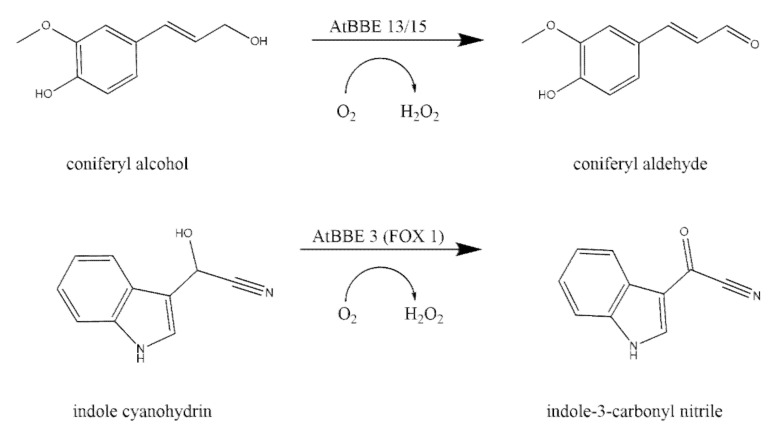
Enzymatic reactions catalyzed by *A. thaliana* (*At*) BBE-like proteins in *A. thaliana* [54].

**Figure 7 ijms-21-05371-f007:**
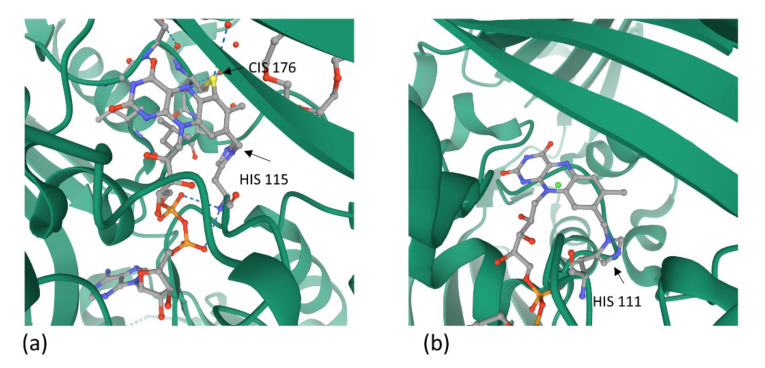
(**a**) Bi-covalent binding of FAD to *At*BBE15 (PDB ID: 4DU8) [58]. (**b**) Mono-covalent binding of FAD to *At*BBE-like 28 (PDB ID5D79) [61]. A part of the to protein structures is always presented in a green ribbon diagram which consists the FAD-binding sites with the contributing amino acid residues (with spheres in grey for C, in red for O, in orange for P, in blue for N and in yellow for S atoms

**Figure 8 ijms-21-05371-f008:**
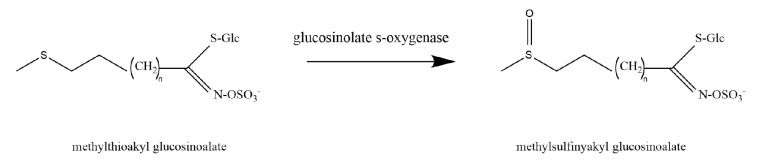
Enzymatic reactions catalyzed by flavin monooxygenase glucosinolate s-oxygenase enzymes (FMO GS-OX) in *A. thaliana.*

**Figure 9 ijms-21-05371-f009:**
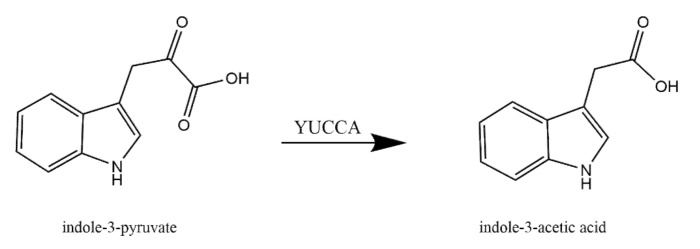
Conversion of indole-3-pryuvate to indole-3-acetic acid catalyzed by indole-3-pyruvate monooxygenase (YUCCA).

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
