# Peer review of "The Flavoproteome of the Model Plant Arabidopsis thaliana"

_ijms, 2020, doi:10.3390/ijms21155371_

Round 1

Reviewer 1 Report

The effort made by the authors is very valuable, as nowadays it is impossible to follow most of the current literature on a topic of interest. This review paper is focused on the potential of flavin mononucleotide and flavin adenine dinucleotide encoded by the genome of Arabidopsis thaliana. The manuscript fits within the scope of the journal. The title is clear and it is adequate to the content of the article. The body of the paper is clear and does not contain major errors. The discussions, conclusions or summary are accurate and supported by the content.

I have some minor recommendations for authors:

- Language style of the paper should be improved

- Please include paternity to Latin scientific name (title, abstract, introduction…)

- L46 …” …Beta vulgaris and sunflower…” For both species choose the scientific or popular name

- L109, L122,  – Please see the error!

Author Response

I have some minor recommendations for authors:

- Language style of the paper should be improved

Response: We tried to improve readability of the manuscript.

- Please include paternity to Latin scientific name (title, abstract, introduction…)

- L46 …” …Beta vulgaris and sunflower…” For both species choose the scientific or popular name

Response: The Latin names of species were added     

- L109, L122,  – Please see the error!

Response: The mistakes were corrected.  

Reviewer 2 Report

The manuscript of Schall et al. reports the inventory of potential FAD/FMN-binding proteins that consists the flavoproteome of plant model species Arabidopsis thaliana. The authors define three criteria to identify and select the flavoenzymes. Most flavoproteins bind FAD contrary to few that bind FMN, whereas the minority bind both FAD and FMN. The flavoproteins were classified in terms of their subcellular localization, enzymatic activity and biological process. The review article sheds light on the function of plant flavoproteins associated with cell wall lignification, response to environmental stresses, biosynthesis of glucosinolates and hormone metabolism. While this collection of flavin-binding proteins might not be sufficient, it is the first report of the flavoproteome of plants, after these of yeast and human, and definitely attracts much of attention.

There are just few points, mainly about the criteria of flavoproteins selection that should be thoroughly addressed for clarity and better understanding by the readers.

#1. Table S1: It would be beneficial for the article to split this table in three parts based on the three selection criteria. Doing so, the data will be easily understood by the readers.

#2. Table S1: Given that a part of flavoproteins was identified by experimental work and the second part was selected based on literature information, the references should be included in the supplemental material of the first two parts of the new Table S1.

#3. Lines 188-193 and Table S1: While the third part of flavoproteins were identified by structural analysis, there is need for more information about potential motifs, structural signatures or domains that are evolutionary conserved enough for the classification of Arabidopsis flavoproteins. Towards this direction the authors are advised to present Multiple Sequence Alignments of the proteins or show the structure of highly conserved domains by utilizing 3-D prediction models.

#4. Results and Materials and Methods sections: It is not clear how the flavoenzymes were grouped depending on their subcellular localization. The authors should provide the references and methodologies for this task.

#5. Figure 5. Both the content and style of this Figure can be improved by outlining clearly the enriched GO terms and their categories. A suggestion could be the use of a two-way bar chart using log10 p-values and percentages of genes associated to a GO-term.

#6. Lines 433-434. Please clarify why the term “flavin mononucleotide binding” was not used in the dataset.

#7. Line 443. p<0.5 or p<0.05?

Author Response

 There are just few points, mainly about the criteria of flavoproteins selection that should be thoroughly addressed for clarity and better understanding by the readers.

 #1.Table S1: It would be beneficial for the article to split this table in three parts based on the three selection criteria. Doing so, the data will be easily understood by the readers.

Respponse: According to the request of the edior, the table S1 was entirely modified divides into three parts, in which the flavin proteins were sorted and grouped persuant to the evidence codes 1, 2 and 3.

#2.Table S1: Given that a part of flavoproteins was identified by experimental work and the second part was selected based on literature information, the references should be included in the supplemental material of the first two parts of the new Table S1.

Response : All references were added for each of the falvoproteins

#3.Lines 188-193 and Table S1: While the third part of flavoproteins were identified by structural analysis, there is need for more information about potential motifs, structural signatures or domains that are evolutionary conserved enough for the classification of Arabidopsis flavoproteins. Towards this direction the authors are advised to present Multiple Sequence Alignments of the proteins or show the structure of highly conserved domains by utilizing 3-D prediction models.

Response: We emphasize here again that the compilation of proteins belonging to the evidence code 3 does not base on structural analysis, but based on data bank and protein sequences (see line 94-96).

In line 188-193, analysis of the structure clans of the entire flavo proteome was described. This analysis bases on protein sequences

 To comply with the advices of the reviewer an additional table (S4) was created that highlights the analysis of the predicted 3-D structure of all 38 proteins collected in category 3 (Table S4., Pyre structure prediction), Two further tabs of Table S4 show the results obtained by structure analysis according to HMMER analysis as well as the the new matches of the search results obtained by the Uniprot. data base. All predicted structures, were collected in  a zip file (Supplements pdb structure prediction evidence code 3)

#4. Results and Materials and Methods sections: It is not clear how the flavoenzymes were grouped depending on their subcellular localization. The authors should provide the references and methodologies for this task.

Response: We followed the advice of the reviewer.

#5. Figure 5. Both the content and style of this Figure can be improved by outlining clearly the enriched GO terms and their categories. A suggestion could be the use of a two-way bar chart using log10 p-values and percentages of genes associated to a GO-term.

Response: We habe added a new figure, here Figure 5, which points to the twenty portein with the most significant p-values. The detailed list of all results of the GO enrichmnets will be found in the new Table S5)

#6. Lines 433-434. Please clarify why the term “flavin mononucleotide binding” was not used in the dataset.

Response: This is directly explained in the sentences.

#7. Line 443. p<0.5 or p<0.05?

Response: This failure was corrected.

Reviewer 3 Report

Dear Authors,

Congratulations on your paper. I found that the data you presented is well-reported, complete and necessary, as little information is available on FAD and FMN-binding proteins in Arabidopsis. I believe that this information will be especially useful for future research on plant development and response to external hazards. 
In the manuscript, I found only minor English corrections needed, but besides that, I have no further comments to make.

All the Best.

Author Response

Thank you for your generous comment

Round 2

Reviewer 2 Report

The authors successfully addressed the comments that were beneficial for the manuscript.